# The perception of eye contact is associated with men's need to belong, self-esteem, and loneliness

**Manman Zhai** [iD]**, Heini Saarimäki, Jari K. Hietanen**\*

Human Information Processing Laboratory, Faculty of Social Sciences/Psychology, Tampere University, Tampere, Finland

\* jari.hietanen@tuni.fi

## Abstract

The present study (N = 367) investigated the association of the cone of direct gaze (CoDG; a range within which slightly averted gaze is accepted as direct gaze) width with the need to belong (NTB) and the possible further influence of NTB on self-esteem and loneliness. Results showed that: (i) men's CoDG width predicted the NTB, self-esteem, and loneliness; (ii) men's narrow CoDG predicted high loneliness via a high NTB and low self-esteem; (iii) a narrow CoDG was associated with low self-esteem via high loneliness only in high-NTB men. Among women, the CoDG width was not associated with the NTB, self-esteem, and loneliness. The findings are discussed considering the formation and maintenance of relationships together with gender differences.

## 1. Introduction

As social beings, most humans live within an extensive network of relationships that, to a large extent, define their identity and personality. In interpersonal interactions, the direction of others' eye gaze, along with their other facial information (e.g., facial expression), provides important clues indicating the dynamics of relationships, such as initiation, sustenance, or termination, between individuals. The sensitivity to eye gaze, which plays a pivotal role in the development of social cognition, is observed from early infancy [1]. Despite showing a preference for faces with a direct gaze over those with an averted gaze from infancy [2], humans are not flawless in detecting a direct gaze. This has been investigated by measuring the cone of direct gaze (CoDG), a range of gaze directions slightly deviating from the direct gaze which observers accept as eye contact [3] (see section Data analysis for how CoDG width was calculated). The width of CoDG has significant inter-individual differences [3,4]. Some people accept considerable gaze deviations as direct gaze (a wide CoDG), whereas others seem to be more conservative in perceiving whether they are being looked at or not (a narrow CoDG). In the present study, we were interested in investigating if the width of individuals' CoDG would be associated with their socioemotional characteristics. In this respect, individuals' belongingness need is a likely candidate. Individuals with a narrow CoDG may experience high belongingness needs, as perceiving others' gaze direction as averted away from them may elicit the feelings of social exclusion, consequently reducing the sense of belonging [5,6]. The current article aims first to investigate whether the CoDG width predicts the strength of a

**Data availability statement:** The data and all the experimental materials are available in the Open Science Framework, https://osf.io/3dhru/.

**Funding:** This work was supported by China Scholarship Council-Tampere University joint scholarship programme (grant #202106750008 to M.Z.), by the Strategic Research Council (grant #352648, #352655 to H.S.), and by the Research Council of Finland (grant #330158 to J.K.H.). The funding sources have no involvement in study design, data collection, data analysis, data interpretation, writing of the article, or the decision to submit the article for publication.

**Competing interests:** The authors have declared that no competing interests exist.

need to belong (NTB; a need to form and maintain at least a minimum quantity of meaningful interpersonal relationships [7]). Furthermore, as high levels of NTB are closely associated with low self-esteem [8] and high loneliness [7,9], especially among individuals with unsatisfying relationships [10,11], we also investigate whether individual differences in the strength of NTB serve as a mechanism (mediator) and/or a condition (moderator) through which the CoDG width can affect self-esteem and loneliness.

## 1.1. Eye gaze and the need to belong

Eye gaze plays a critical role in regulating interpersonal interactions. Humans can use their own gaze direction to express intimacy and social control [12]. In turn, a social counterpart's gaze direction can be exploited by an individual as a pivotal signal to evaluate the relations between them [6]. Another person looking away (i.e., averted gaze) communicates that a perceiver is not within the focus of the looker's attention. In social communication, not making eye contact conveys disinterest or active avoidance, often as a demonstration of disengaging oneself from another individual. Seeing an averted gaze may render the feelings of being ostracized or socially excluded [5,6,13,14]. The interpretation of an averted gaze as relational devaluation leads observers to infer less positive personality traits about the looker, compared to seeing a person making eye contact. In this way, perceiving an averted gaze may further hinder the initiation and maintenance of relationships. In contrast, a looker making eye contact with the perceiver (i.e., direct gaze) indicates their interest and potential communication intentions. Being looked at means that the self of a perceiver is the target of the looker's attention [15], automatically eliciting positive affective reactions in the perceiver [16]. Seeing a direct gaze compared to an averted gaze also activates the brain systems associated with approach motivation [17,18]. Furthermore, direct gaze facilitates processing of social counterparts' emotions [19], face memory [20], joint attention between individuals [21], and empathy [22]. Previous studies have also provided evidence that direct gaze leads perceivers more likely to give positive appraisals of others [6,23] and exhibit prosocial behaviours [24,25]. Therefore, perceiving another person's direct gaze is beneficial to build and sustain satisfying relationships.

Establishing and maintaining relationships is one of the central aspects of human social life [26]. Baumeister and Leary [7] hypothesized that humans have a fundamental need "to form and maintain at least minimum quantity of interpersonal relationships" (p. 499). Interestingly, this need to belong has been demonstrated to relate also to the perception of eye gaze direction. Merely receiving an averted gaze (vs. a direct gaze) elicits strong feelings of ostracism (i.e., being socially excluded and ignored) and increases the NTB [5,6]. Considering the association between perception of eye gaze and the NTB, in the present study, we aimed to investigate whether people's CoDG width predicts their NTB. As stated earlier, perceiving another individual's gaze as direct could increase the observer's own approach motivation. A wide CoDG, if not taken to the extreme, is beneficial for daily social interactions. Through repeatedly accepting a considerably large range of gaze deviations to be direct gaze, observers can maximally avoid the cost of missing a mutual gaze. Thus, wide-CoDG individuals may have greater possibility for interaction opportunities and, therefore, they may develop a sense of connectedness that reduce their need to seek additional relationships. In contrast, the likelihood of exchanging affiliative cues is smaller for individuals with a narrow CoDG, and this could result in scarce opportunities to initiate and maintain relationships.

Individuals' CoDG width has been suggested to be highly stable over time. A study using eye contact judgment task measured the CoDG width and repeated the measurement after 5 minutes and/or after 1 week. The results demonstrated a high agreement in the CoDG width within participants, indicating that CoDG is a stable personality-like trait [27]. Another study

aiming to identify a neural trait marker for individual differences in the CoDG width revealed that resting theta current density in the left temporo-parietal junction (TPJ) and posterior superior temporal sulcus (pSTS) was associated with the CoDG width. Specifically, higher baseline cortical activation in the left TPJ/pSTS was associated with a wider CoDG [28]. The authors proposed that higher baseline activation in TPJ/pSTS could be associated with higher perspective-taking abilities. Individuals with higher perspective-taking abilities tend to be more liberal in determining whether another person is making eye contact or not, making them more likely to approach someone who is supposedly making eye contact. Therefore, the stability of the CoDG width may contribute to the persistence and immutability of people's approach-avoidance motivation in social interactions.

When investigating the association between CoDG and NTB, gender differences must be taken into consideration. Gender differences play a role in the judgments of being looked at; men are more likely to believe that another person is making eye contact with them compared to women [29,30], and men have a wider CoDG than women [31,32]. Additionally, previous research has shown that women have a greater NTB than men [33], possibly indicating increased affiliative focus among women. The gender difference in NTB may also signify that women tend to construe their self as connected to others (interdependent self-construal), whereas men highlight their uniqueness in self-construal (independent self-construal) [34]. Thus, it is interesting to know whether the associations between CoDG and NTB vary depending on participants' gender.

## 1.2. The joint effect of eye gaze and NTB on self-esteem and loneliness

Low self-esteem and high loneliness are indicators of poor-quality social connections, and they have been shown to be related to high levels of NTB [35]. Given that the NTB is such a powerful and fundamental motivation to form social relationships [7], people need internal mechanisms to gauge whether it remains at an acceptable level. According to the sociometer theory, self-esteem is akin to an internal meter functioning to monitor the quality of an individual's interpersonal relationships [8]. High self-esteem reflects that an individual feels to be a valued and desirable person in their groups and close relationships. Contrarily, low self-esteem indicates that the current level of social acceptance and inclusion is not satisfying. Moreover, the more an individual cares about social connectedness with others, the more their self-esteem is dependent upon their NTB level. The negative feelings associated with loneliness arise from a discrepancy between individuals' desired and achieved social connections [36]. The NTB has been proposed to have two main features: people need frequent personal contacts with other people, and people need to perceive the stability and affective concern in their interpersonal bonds [7]. Having meaningful and supportive social bonds but lacking frequent personal interactions could lead to loneliness. For example, compared to nonlonely people, lonely people spend less time with friends and family members who are most likely to satisfy one's NTB [37]. Likewise, frequent contacts with non-supportive and indifferent others would do little to satisfy the NTB. Loneliness has been shown to be more a matter of a lack of intimate connectedness than a lack of social contacts [38,39].

In addition to NTB, the CoDG width may also predict the levels of self-esteem and loneliness. Compared to receiving a direct gaze, a person receiving another's averted gaze experiences a reduction in their self-esteem [6]. The developmental history of evaluating relationships based on evaluating how frequently, in general, one is being looked at can predict the level of self-esteem over the long run. Thus, it is possible that individuals with a stable narrow CoDG width tend to have lower self-esteem. Additionally, if without opportunities for re-/inclusion in interaction, an individual may withdraw and seek solitude [40,41]. Therefore, an individual's narrow CoDG width may contribute to their loneliness, as narrowed CoDG

renders one to miss potential interaction opportunities or feel that the relationships with others are depreciated.

In sum, NTB may be a mechanism (a mediator) through which the CoDG width affects the levels of self-esteem and loneliness; a high NTB could potentially decrease self-esteem and increase loneliness. Numerous cross-sectional studies have documented a strong association between self-esteem and loneliness (for a meta-analysis, see [42]). Thus, the association may indicate that low self-esteem is a risk factor for experiencing high loneliness [43,44], or that feeling lonely reduces the level of self-esteem [45,46]. Whatever the direction of the causality between these two is, low self-esteem and high loneliness reciprocally reinforce one another and severely affect people's physical and psychological well-being.

It is also possible that the level of NTB moderates the effect of the CoDG width on self-esteem and loneliness. Previous research has emphasized the moderating effect of the NTB in studies investigating the effects of negative factors in social interaction on psychological problems and maladaptive behaviours. For example, negative interpersonal interactions as well as social exclusion were more strongly associated with suicidal ideation and alcohol consumption in high-NTB individuals compared to low-NTB individuals [47–49]. Therefore, in the present study, a low NTB may buffer the effect of the narrow CoDG width on self-esteem and loneliness.

### 1.3.  Overview of the present study

The primary research question was whether CoDG predicts NTB, self-esteem, and loneliness. We hypothesized that individuals with a narrow CoDG may have a high NTB, low self-esteem, and high loneliness. In addition, based on various perspectives in the research literature, we also investigated whether CoDG predicts self-esteem and loneliness via NTB (serial mediation models, Fig 1a and 1b; moderated mediation models, Fig 1c and 1d). In the present study, we first investigated the relationship between CoDG and NTB. Because previous research has shown that there are gender differences in the CoDG width [31,32] as well as in NTB strength [33,50,51], we also explored whether there were gender differences in NTB and CoDG in the present sample of participants. If that would be the case, we would analyse the data separately for men and women.

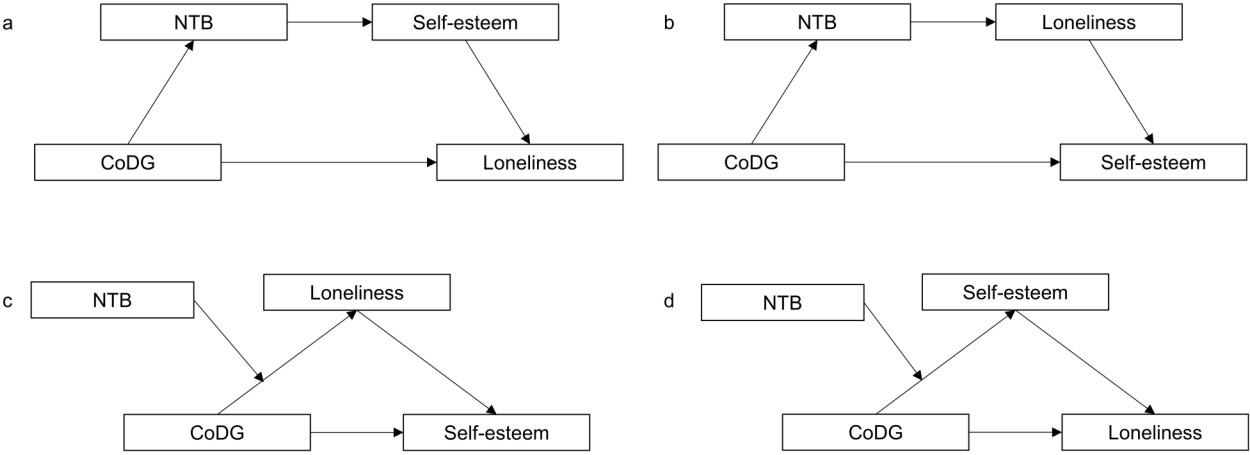

**Fig 1.  Serial mediation models (a, b) and moderated mediation models (c, d).**

To this end, in an online study, participants were required to complete an eye contact judgment task in which they judged whether a neutral face was looking at them or not (a direct gaze or an averted gaze slightly deviated from the direct gaze at steps of 2˚, 4˚, 6˚, or 8˚). Following the eye contact judgment task, participants completed NTBS [35], Rosenberg Self-Esteem Scale [52], and UCLA Loneliness Scale [53].

## 2. Methods

### 2.1. Participants

The present study recruited 400 adult participants (average age: 38.31 years; range: 18-78 years; recruitment period: 25/04/2023 – 02/05/2023) online via Prolific (https://www.prolific. co/) [54]. Participants were included if a) they lived in the United Kingdom, b) their first language was English, and c) their Prolific approval rate was 100%. The participants were also required to have a normal or corrected-to-normal vision and no neurological or psychological disorders. Additionally, participants who had taken part in other eye contact judgment tasks by the present authors were blocked. Participants were strongly encouraged to perform the tasks using a desktop computer, but the ones who used laptops could not be filtered out. The principles outlined in the Declaration of Helsinki were followed when conducting this research. The Ethics Committee of Tampere region approved that the present study was exempt from a formal ethical review and evaluated that the present study followed good scientific practice. The link to the present study was not accessible to participants who used tablets or smartphones. The data and all the experimental materials are available in the Open Science Framework, https://osf.io/3dhru/.

### 2.2. Stimuli and programming of the experiment

The face stimuli were four neutral faces (two females and two males) which were created using 3D animation software (Digital Art Zone 3D studio, Daz 3D) (https://www.daz3d.com/), thus the face images did not depict real people. The gaze direction varied in steps of 2˚ from direct gaze (0˚) to leftward and rightward averted gaze (2˚, 4˚, 6˚, 8˚). For attention check purpose, gaze stimuli of 20˚ were also included (see below). Mirror images of the face stimuli were created by flipping the original images horizontally to avoid the potential influence of facial asymmetry. An oval area of the face region was cropped to eliminate the possible influence of hair and ears (see Fig 2a). Each face model was evaluated equally attractive by women and men in another stimulus assessment study. See S1 Table in Supplemental file for the attractiveness ratings and corresponding statistics.

Lab.js (https://lab.js.org/), a free, open, and online study builder, was used to program the experimental tasks [55]. The study was uploaded to Open Lab (https://open-lab.online/) [56] to generate an external link which was inserted on Prolific, the platform used to recruit participants.

### 2.3. Procedure and measures

The invitation of the study contained the aim and duration of the study, a brief description of what the participants were required to do in the study, the abovementioned prerequisites for participation, and emphasis of the confidential and voluntary nature of their participation. They were told that by clicking the external link to start the study meant that they gave informed consent to participate in the study and that the results and anonymized data could be published.

The participants were required to adjust the distance between them and the screen at full arm's length and ensure they were seated directly in front of the computer screen. After the

position adjustment, they completed the tasks in a fixed sequence: the eye contact judgment task and the questionnaire including the Need to Belong Scale, Rosenberg Self-Esteem Scale, and UCLA Loneliness Scale. Before each task, detailed information was presented to guide them on how to perform the tasks.

**2.3.1. Eye contact judgment task.** The eye contact judgment task consisted of two blocks between which participants could have a self-paced break. In each block, two female and two male identities with each averted gaze direction (left and right: 2˚, 4˚, 6˚, and 8˚) were presented twice, resulting in 64 trials. The gaze direction of 0˚ was presented four times for each identity resulting in 16 trials. In addition, we also included faces with the gaze direction of 20˚ (left and right) to be used as attention check. These stimuli were presented once for each identity resulting in 8 trials. Thus, in total, there were 88 trials in each block. Each trial started with an 850-ms fixation followed by a face picture for 300 ms. Subsequently, a blank screen was presented for 500 ms, followed by a response screen waiting for participants' judgment of whether "the person in the picture is or is not looking directly at you". On the response screen, the word "yes" appeared in the upper part of the screen, while the word "no" appeared in the lower part of the screen. Correspondingly, participants pressed "y" on their keyboards if they felt that the person was looking at them and they pressed "n" if they felt that the person was not looking at them (see Fig 2b).

**2.3.2. The Need to Belong Scale (NTBS).** To assess individual differences in the desire for acceptance and belonging, NTBS [35] was used. The scale required participants to indicate the extent to which they agreed on ten statements using a 5-point rating scale ranging from 1 (strongly disagree) to 5 (strongly agree). The statements were presented one by one, in a random order, and the 5-point scale was shown below each statement. Among the 10 statements, one attention check item was inserted: "*Please press number key five to show you are paying attention to this question*".

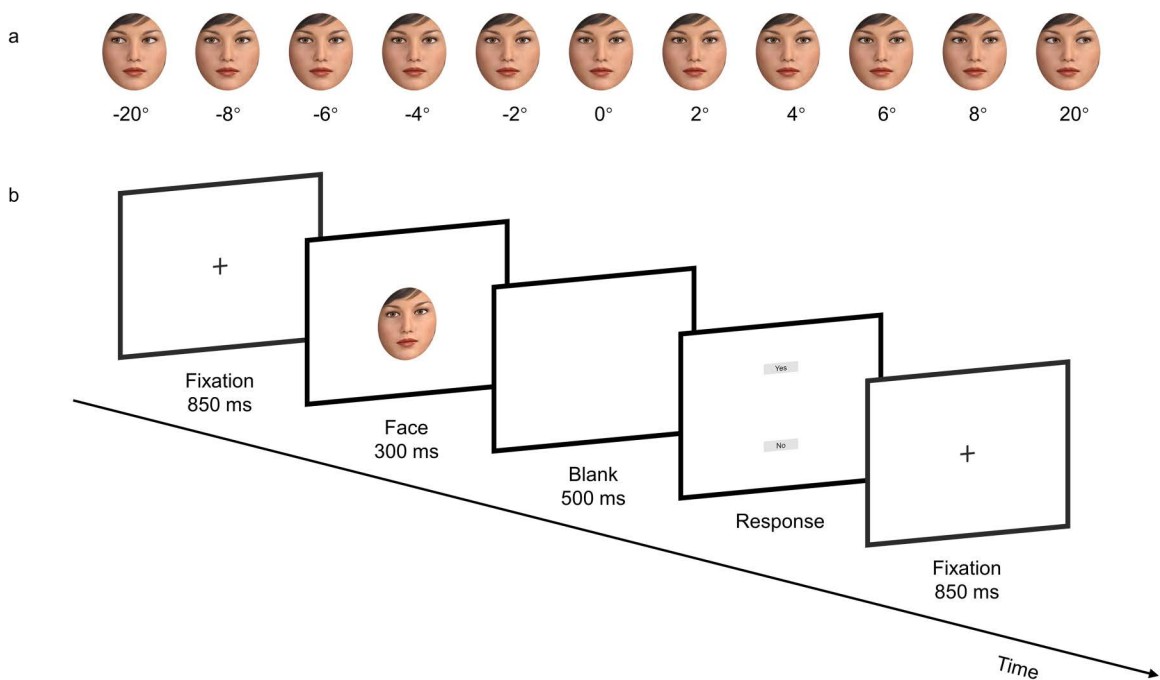

**Fig 2. Experimental stimuli (a) and schematic flow of a trial (b).**

**2.3.3.  Rosenberg Self-Esteem Scale.**  To assess an individual's self-esteem, participants had to indicate the extent to which they agreed on ten items of the Rosenberg Self-Esteem Scale (RSE; [52]) using a 5-point scale ranging from 1 (strongly disagree) to 5 (strongly agree). The statements were presented one by one in a random order, and the 5-point scale was shown below each statement. Among the 10 statements, one attention check item was inserted: "*Please press number key five to show you are paying attention to this question*".

**2.3.4.  UCLA Loneliness Scale.**  To measure the feeling of being disconnected from others (loneliness), participants had to answer 20 questions from UCLA Loneliness Scale [53] using a 5-point scale ranging from 1 (never) to 5 (always). The questions were presented one by one in a random order, and the 5-point scale was shown below each statement. Among the 20 questions, two attention check questions also were inserted in this section, "*How often do you answer these kinds of surveys? Press number key one to show you are paying attention to this question.*" and "*How often do you spend your time on the internet? Press number key one to show you are paying attention to this question.*"

## 2.4.  Data analysis

Complete data from four hundred participants were collected. To exclude participants who did not pay enough attention to the tasks, participants' answers to the attention check items were first inspected. Their responses to attention check items determined whether or not they would get paid. In the eye contact judgment task, the participant was excluded if the percentage of "yes" responses to catch trials (faces with gaze direction of 20°) was more than 50%. One participant who gave 87.5% "yes" responses to these catch trials was excluded. In the questionnaire (composed of 40 items), participants were excluded if they failed more than two (out of four) attention check trials. No participant was excluded due to attention checks in the questionnaire (19 participants failed one attention check question, and 2 failed two). Thus, 399 participants' data were analyzed subsequently. In the subsequent analysis of gaze cone for the 399 participants, those whose data did not meet the criteria for calculating CoDG width were excluded, even if they had passed the attention check questions among the three scales.

Before the calculation of the CoDG width for every participant, two criteria were used to exclude participants. The participant was excluded if the percentage of "yes" responses to the gaze direction of 8° (leftward and rightward averted gaze) was more than 50% or the percentage of "yes" responses to direct gaze (0°) was less than 50%. Based on the criteria, 11 participants were excluded. 2 participants reporting technical issues that could have influenced their results were also excluded. Naturally, the 20° gaze trials were removed from the subsequent data analysis.

The data of the eye contact judgment task were analyzed by calculating a point of subjective equality (PSE; the point at which the gaze direction has a 50% probability to be judged as a direct gaze or averted gaze) for each participant. The width of CoDG was approximated as the PSE multiplied by two (to cover both the left and right sides). The PSE was calculated using a binary logistic regression model for each participant. The calculation was conducted in R using the *glm* function and *logit* link function, in which the gaze direction (0°/2°/4°/6°/8°) was the independent variable, and the dichotomous responses to whether being looked at or not was the dependent variable. Following the formula below, the PSE equals $-\beta_0/\beta_1$ with a P value (the probability of a "yes" response) of 50%.

$$\log\left(\frac{P}{1-P}\right) = \beta_0 + \beta_1 x$$

After the calculation of the CoDG width, the goodness of model fit for the binary logistic regression models was assessed by examining the deviance residuals. A deviance residual higher than the corresponding degree of freedom indicates that the model is a poor fit for the data. Based on this criterion, another 19 participants were excluded. In sum, the final data analysis from the eye contact judgment task was based on the data from 367 participants (178 men: $M_{Age}$ = 37.53, $SD$ = 13.12; 189 women: $M_{Age}$ = 37.95, $SD$ = 12.83; There was no age difference between men and women: $t(365)$ = 0.31, $p$ = 0.760). We also checked these participants' responses on attention check trials (16 trials of 20˚ gaze direction), and the average error rate on the catch trials was 0.46%. Two participants made 2 erroneous responses, twenty-three participants made 1, and the remaining 342 participants made no erroneous responses. Till here, we reported all manipulations, measures, and exclusions in the present study.

For the questionnaire (three scales), attention check items were first removed. Then, the scores for all the items were summed up for each scale (following reverse scoring for some items). High scores on NTBS indicate a high need to belong. High scores on the Rosenberg Self-Esteem Scale indicate high self-esteem. High scores on UCLA Loneliness Scale indicate high loneliness.

Statistical analyses were conducted using SPSS for Windows statistical package (IBM SPSS Statistics, Version: 28.0). Normality tests were conducted for the whole data as well as for men and women separately before all analyses. Variables that did not follow normal distribution were log10 transformed to achieve normal distribution. However, the results were the same when based on transformed and untransformed data. Therefore, the data were finally analyzed in their original form because the sample size was large enough to reduce the impact of any skewness/kurtosis. Independent samples $t$-tests were conducted to investigate the effects of gender on CoDG, NTB, self-esteem, and loneliness. Pearson product-moment correlations were conducted for all the variables. The mediation and moderation analyses were carried out with PROCESS v4.2 developed for SPSS by Hayes [57]. PROCESS model 6 was used to determine the serial mediation effects in serial mediation models (Fig 1a and 1b) and PROCESS model 7 was used to determine the moderated mediation effects in the moderated mediation models (Fig 1c and 1d). The 95% confidence interval was used, and the number of bootstrap samples was 5000. The calculated 95% confidence intervals should not contain zero to indicate that the mediating effect obtained from the bootstrap process was significant [58].

## 3. Results

### 3.1. The effect of gender on the width of CoDG, NTB, self-esteem, and loneliness

The results of the independent samples $t$-tests revealed significant effects of gender on the width of CoDG and on NTB. Men had a wider gaze cone ($M$ = 8.96˚, $SD$ = 2.45˚) than women ($M$ = 8.40˚, $SD$ = 2.35˚; $t(365)$ = 2.26, $p$ = 0.025, $d$ = 0.24, 95% CI [0.07, 1.06]). Women, in turn, showed a greater NTB ($M$ = 33.43, $SD$ = 7.22) than men ($M$ = 29.27, $SD$ = 6.55; $t(365)$ = 5.77, $p$ < 0.001, $d$ = 0.60, 95% CI [2.74, 5.58]). There were no significant differences between men and women in self-esteem and loneliness (all $ps$ > 0.05; see the right part of Table 1 for all the statistics of independent samples $t$-tests).

### 3.2. The correlations between the width of CoDG, NTB, self-esteem, and loneliness

Because of the effects of gender differences on the width of CoDG and NTB, the results of correlational analyses are reported separately for men and women (see Table 2). For men, the width of CoDG, NTB, self-esteem, and loneliness were correlated with each other (CoDG &

**Table 1.** Left part: Means and standard deviations of the width of the cone of direct gaze (CoDG), the need to belong (NTB), self-esteem, and loneliness for all, men, and women; Right part: Statistics for independent samples *t* test.

| | All (N = 367) | | Men (N = 178) | | Women (N = 189) | | The effects of participants' gender | | | |
|---|---|---|---|---|---|---|---|---|---|---|
| | Mean | SD | Mean | SD | Mean | SD | t | p | Mean Difference | 95% CI of the Difference |
| CoDG | 8.67 | 2.41 | 8.96 | 2.45 | 8.40 | 2.35 | 2.26 | 0.03 | 0.56 | [0.07, 1.06] |
| NTB | 31.41 | 7.20 | 29.27 | 6.55 | 33.43 | 7.22 | −5.77 | < 0.001 | −4.16 | [−5.58, −2.74] |
| Self-esteem | 34.84 | 8.65 | 34.73 | 9.06 | 34.95 | 8.27 | −0.24 | 0.81 | −0.22 | [−2.00, 1.56] |
| Loneliness | 54.76 | 14.62 | 55.97 | 14.80 | 53.62 | 14.39 | 1.54 | 0.13 | 2.34 | [−0.66, 5.34] |

**Table 2.** Correlations between CoDG, NTB, self-esteem, and loneliness for men and women.

| Men (N = 178) | | | | | Women (N = 189) | | | | |
|---|---|---|---|---|---|---|---|---|---|
| | CoDG | NTB | Self-esteem | Loneliness | | CoDG | NTB | Self-esteem | Loneliness |
| CoDG | 1 | | | | CoDG | 1 | | | |
| NTB | −.15* | 1 | | | NTB | 0.03 | 1 | | |
| Self-esteem | .16* | −.26** | 1 | | Self-esteem | −0.07 | −.29** | 1 | |
| Loneliness | −.16* | .19* | −.69** | 1 | Loneliness | 0.11 | 0.01 | −.58** | 1 |

** Correlation is significant at the 0.01 level (2-tailed).

* Correlation is significant at the 0.05 level (2-tailed).

NTB: $r$ = -0.15, $p$ = 0.049; CoDG & self-esteem: $r$ = 0.16, $p$ = 0.033; CoDG & loneliness: $r$ = -0.16, $p$ = 0.030; NTB & self-esteem: $r$ = -0.26, $p$ < 0.001; NTB & loneliness: $r$ = 0.19, $p$ = 0.011; self-esteem & loneliness: $r$ = -0.69, $p$ < 0.001). For women, CoDG was not correlated with NTB, self-esteem, or loneliness (CoDG & NTB; CoDG & self-esteem; CoDG & loneliness; NTB & loneliness; all $p$s > 0.05). However, NTB and self-esteem were negatively correlated, showing that the greater the NTB, the lower was the self-esteem in women ($r$ = -0.29, $p$ < 0.001). Self-esteem was negatively correlated with loneliness, suggesting that lonelier women had lower self-esteem ($r$ = -0.58, $p$ < 0.001).

### 3.3. Mediation and moderation analyses

**3.3.1. Men.** *Serial mediation analyses*: Two serial mediation models (Fig 1a and 1b) were proposed to advance the understanding of the relationship among the variables. In the serial mediation model investigating the effect of CoDG width on loneliness (Fig 1a), three indirect effects, i.e., the indirect effects of CoDG width on loneliness through NTB, self-esteem, and the link of NTB-self-esteem, were postulated to exist. Additionally, we also investigated whether there is an indirect effect of CoDG width on self-esteem through NTB. The significant indirect effects are shown in Fig 3a. The indirect effect of the width of CoDG on loneliness through the link of NTB-self-esteem was significant (indirect effect = a1*d*b2 = -0.15, boot *SE* = 0.09, 95% CI [-0.36, -0.00]). Although the results revealed a mediating effect of NTB on the association between the width of CoDG and self-esteem (indirect effect = a1*d = 0.13, boot *SE* = 0.08, 95% CI [0.00, 0.31]), the indirect effects between CoDG and loneliness through NTB (indirect effect = a1*a2 = -0.01, boot *SE* = 0.06, 95% CI [-0.14, 0.11]) and through self-esteem alone (indirect effect = b1*b2 = -0.51, boot *SE* = 0.32, 95% CI [-1.16, 0.10]) were not significant (see Fig 3a). The direct effect of the CoDG width on loneliness was negative but not statistically significant (direct effect = c' = -0.32, $t$(176) = -0.94, $p$ = 0.347, 95% CI [-0.98, 0.35]). Therefore, the relationship between CoDG width and loneliness was totally dependent on NTB and self-esteem. Additionally, in the serial mediation model investigating the effect of CoDG width on self-esteem (Fig 1b), the three indirect effects

**Fig 3. Results of the significant models testing (a) the serial mediation of NTB and self-esteem on the association between CoDG and loneliness; (b) the associations of CoDG with self-esteem via loneliness at high and low levels of NTB.**

of CoDG width on self-esteem via NTB, loneliness, and the link of NTB-loneliness were not significant; all 95% CIs included zero.

*Moderated mediation analyses*: Two moderated mediation models (Fig 1c and 1d) were proposed. The moderated mediation model investigating the effect of CoDG width on loneliness (Fig 1c) did not show the moderating effect of NTB on the indirect effect of CoDG width on loneliness via self-esteem, as the index of moderated mediation did not reach significance ($b$ = -0.04, boot $SE$ = 0.05, 95% CI [-0.13, 0.05]) [57]. In the other moderated mediation model investigating the effect of CoDG width on self-esteem (Fig 1d), the index of moderated mediation was significant ($b$ = 0.06, boot $SE$ = 0.03, 95% CI [0.00, 0.11]). The slope analyses demonstrated that narrow CoDG width predicted low self-esteem through high loneliness only for those men who had higher NTB (+1 SD: $b$ = 0.77; boot $SE$ = 0.30; 95% CI [0.14, 1.34]), but not for those who had lower NTB (-1 SD: $b$ = 0.04; boot $SE$ = 0.23; 95% CI [-0.46, 0.44]).

**3.3.2. Women.** *Serial mediation analyses*: In the serial mediation model investigating the effect of CoDG width on loneliness (Fig 1a), the three indirect effects of CoDG width on loneliness via NTB, self-esteem, and the link of NTB-self-esteem were not significant; all 95% CIs included zero. Also, in the other serial mediation model investigating the effect of CoDG width on self-esteem (Fig 1b), the three indirect effects of CoDG width on self-esteem via NTB, loneliness, and the link of NTB-loneliness were not significant; all 95% CIs included zero.

*Moderated mediation analyses*: The moderated mediation models investigating the effect of CoDG width on loneliness (Fig 1c) and the effect of CoDG width on self-esteem (Fig 1d) did not show the moderating effect of NTB on the indirect effect of CoDG width on loneliness via self-esteem (the index of moderated mediation: $b$ = 0.05, boot $SE$ = 0.04, 95% CI [-0.03, 0.13]) or the moderating effect of NTB on the indirect effect of CoDG width on self-esteem via loneliness (the index of moderated mediation: $b$ = -0.02, boot $SE$ = 0.02, 95% CI [-0.08, 0.02]).

## 4. Discussion

There were two aims in the present study. The primary aim was to examine whether CoDG (the range of gaze directions that are perceived as making eye contact) associates with NTB (the need to form and maintain social bonds), self-esteem, and loneliness. Provided that associations between these variables would be found, we aimed to investigate whether NTB has a role in moderating and/or mediating the relationship between CoDG as a predictor and self-esteem and loneliness as outcomes. Participants were required to complete an eye contact judgment task in which they judged whether they were being looked at or not by a stimulus face. Following this task, they filled in the Need to Belong Scale (NTBS), Rosenberg's Self-esteem Scale, and UCLA Loneliness Scale.

We hypothesized that: (1) The CoDG width predicts NTB, showing that a narrower CoDG is associated with a higher NTB; (2) The CoDG width predicts self-esteem, showing that a narrower CoDG is associated with lower self-esteem; (3) The CoDG width predicts loneliness, showing that a narrower CoDG is associated with higher loneliness. Further, we explored how the CoDG width predicts individuals' self-esteem and loneliness via individual differences in NTB (Fig 1a–1d). The effects of gender differences were considered when examining the hypotheses, as previous research has shown that men have a wider CoDG than women while women have a higher NTB than men.

The results showed that participants' gender had an effect on their CoDG width and NTB. Consistent with previous research, men had a wider CoDG than women [31,32]. We want to point out here that each face stimulus (two male and two female models) used in the present study was evaluated as equally attractive by men and women. This excludes the possibility that the gender differences in the CoDG width would have been caused by greater facial attractiveness of the opposite gender stimulus faces for men. Regarding the effect of gender on NTB, the results showed that women had a greater NTB than men. This is also in accordance with some previous studies [33,50,51]. However, the research findings are not consistent. Leary et al. [35] conducted a series of nine studies examining the construct validity of NTBS and reported (footnote 2, p. 622) that in about half of the nine studies men scored significantly higher than women, while in the remainder there was no difference between men and women. Baumeister and Sommer [59] proposed that both men and women pursue belongingness and that if there are any gender differences in NTB, they are reflected on which spheres of belongingness men and women pursue and which strategies and criteria they use in this pursuit. For example, to satisfy the NTB, women orient toward intimate and close relationships while men orient toward superficial and large groups. We do not attempt to interpret the gender differences in CoDG and NTB, at this point, but only note that since we found gender differences in these variables, we explored the relationships among CoDG, NTB, self-esteem, and loneliness separately for men and women. In the following, we discuss the results separately for men and women.

The results showed that, among men, the CoDG width predicted the levels of NTB, self-esteem, and loneliness; a narrower CoDG was associated with a higher NTB, lower self-esteem, and higher loneliness. Given that an averted gaze has been interpreted to indicate the sender's avoidance-oriented motivation [60,61] and activate a receiver's motivational tendency of avoidance [17,18], a narrow CoDG may negatively impact the formation and maintenance of social bonds. Thus, men with a narrower CoDG might exhibit higher levels of NTB compared to those with a wide CoDG. Additionally, the correlation between CoDG and self-esteem may result from that men with a narrow CoDG may need to struggle to initiate desired relationships and perceive that their existing relationships are being threatened, and this leads to low self-esteem. The association between CoDG and loneliness may reflect the effect of the narrower CoDG on increasing the difficulty in initiating or sustaining desired social relationships. This lack of meaningful social bonds or difficulty in maintaining existing social relationships can lead to social isolation.

Considering that the present study is a correlational study, cautiousness is warranted when interpreting the results by assuming causation. When considering the direction of influence between the variables, it is possible that the associations between CoDG and self-esteem and between CoDG and loneliness result from the influences of men's self-esteem and loneliness on their perceptions of whether or not they are being looked at by others. According to the sociometer theory of self-esteem [8], self-esteem not only monitors the relational value with other people in immediate interpersonal context but also assesses the general acceptability by others across various situations and over time [62]. Indeed, self-esteem correlates highly

with individual's current beliefs about the degree to which they are generally accepted, valued, and supported by others [63,64]. Therefore, it is not surprising that individuals with low self-esteem, in contrast to those with high self-esteem, perceive fewer opportunities to be accepted and included. Consequently, men with low self-esteem could tend to judge that they are less likely to be looked at by others, resulting in a narrow CoDG. A narrow CoDG can also be anticipated among lonely men. According to the model of loneliness [65,66], loneliness triggers hypervigilance for social threat, biasing lonely individuals to perceive the social world as more threatening and anticipate more negative social interactions compared to their nonlonely counterparts [67,68]. Being reluctant to be at risk for rejection, lonely men may not interpret slight gaze deviation as inviting signals directed toward them. Thus, a high level of loneliness could predict a narrow CoDG.

However, the alternative reasoning presented above cannot well explain why a higher NTB was associated with a narrower CoDG. We would expect that men with a strong need for social acceptance and connectedness should have a wider CoDG, as perceiving slight gaze deviations as direct gaze could help fulfil this need. Yet, our study found the opposite pattern: a higher NTB was associated with a narrower CoDG. This pattern aligns with previous research suggesting that individuals with a high NTB tend to be more attuned to social information, resulting in improved sensitivity to social cues and elevated accuracy in deciphering the meaning. For instance, studies have shown that those high in NTB are accurate in identifying extroverted faces [69] and inferring others' thoughts and feelings in an empathic accuracy task [70]. Although being sensitive to social cues and accurate in interpreting their meaning can be adaptive, a narrower CoDG does not seem to fulfil the belongingness need, as evidenced by its association with lower self-esteem and higher loneliness in our study. Therefore, considering that people often use gaze direction to assess their connectedness with others [6,14], it is more plausible that the CoDG width predicts the NTB level, rather than the other way around.

The present results from men also revealed that CoDG predicted loneliness through the serial mediation by NTB and self-esteem. The results showed that men with a narrower CoDG reported higher levels of NTB, which, in turn, was associated with lower self-esteem. This reduction in self-esteem was, in turn, associated with heightened feelings of loneliness. This finding indicated that the narrowed CoDG potentially decreases the opportunities to form and maintain relationships among men. Not having either regular contact or an ongoing bond with others cannot fully satisfy NTB. Further, a high NTB may increase concerns regarding relationship devaluation or may lead to the eventuality of unsatisfying relationships which is reflected on low self-esteem. Therefore, men with a narrow CoDG eventually experience that there is a gap between the current and the desired relationships, and loneliness ensues. While the mediating effect of NTB between CoDG and self-esteem was significant, it was observed that neither NTB nor self-esteem alone showed a mediating effect between CoDG and loneliness. It was the serial mediation by NTB and self-esteem that collectively predicted higher levels of loneliness associated with a narrower CoDG. This finding supports the sociometer theory of self-esteem, as the link between NTB and self-esteem plausibly explains why human beings need self-esteem to monitor the quality of their relationships with other people, and why low self-esteem is associated with many problems in social life [8].

The results of the moderated mediation model showed that lower self-esteem was predicted by loneliness, which was in turn influenced by the joint effects of the CoDG width and the strength of NTB. Specifically, a narrower CoDG was associated with higher loneliness in men with a high NTB, but not in those with a low NTB. Further, narrow-CoDG men with a higher NTB experienced lower self-esteem, whereas those with a lower NTB did not. High-NTB individuals are more susceptible to negative effects in adverse social interactions, reporting

lower self-esteem and higher loneliness [71]. However, high levels of NTB are not inherently detrimental and only become so when combined with limited social interaction opportunities [10]. Consistent with this proposal, our findings revealed that the adverse effect of a narrow CoDG on loneliness and self-esteem only affects high-NTB men, not low-NTB men. Parallel to the findings of the present study, individuals with a high NTB but reporting high relationship satisfaction have been reported not to experience high loneliness or low self-esteem [11]. Similarly, belonging to a majority group or practicing self-affirmation can mitigate social exclusion effects for high-NTB individuals [71,72]. Therefore, together with previous research, the present study also points to the avenue to alleviate the adverse effect of negative interpretation of social cues by high-NTB individuals. Establishing an affiliative and inclusionary network to fulfil high NTB would help boost self-esteem and decrease loneliness.

In terms of the present results for women, CoDG was not correlated with NTB, self-esteem, and loneliness, and NTB was not correlated with loneliness. However, similar to men, the results showed a negative association between NTB and self-esteem in women. Additionally, self-esteem was strongly and negatively associated with loneliness. We assumed that the joint effects of CoDG and NTB predicted individuals' self-esteem and loneliness levels. Among women, due to the lack of association between CoDG and NTB, neither serial mediation models nor moderated mediation models were significant. So, how could we explain the gender difference in the association between CoDG and NTB? Baumeister and Sommer [59] proposed, based on the premise that humans, regardless of gender, are fundamentally motivated to establish and sustain relationships with others, that men are oriented to large groups of shallow relationship, while women emphasize dyadic, close, and intimate relationships. Therefore, a possible explanation is that if women's sociality is mainly oriented toward intimate and close relationships, the judgments of being looked at or not by nonsignificant others may not be related to their sense of belonging. Moreover, gender differences in mental images of social groups may also explain why CoDG was correlated with NTB in men but not in women. Men tend to think their groups as categorical (i.e., family or college) while women tend to think of their groups as dyadic bonds (e.g., sister or classmate) [73,74]. The influence of the CoDG width on social interactions may differ depending on the cognitive representation of social groups. For men, seeing groups as overall categories rather than consisting of multiple individuals would make it difficult to form dyadic and meaningful social interactions, which may be exacerbated by a narrow CoDG. Therefore, compared to those with a wide CoDG, men with a narrow CoDG are less likely to have fully satisfied NTB. However, women tend to recognize potential dyadic relationships first and engage in meaningful interactions with specific individuals. Thus, they are more likely than men to have social interactions of high quality to satisfy their NTB, regardless of the CoDG width.

## 5. Limitations

The findings of our study should be interpreted in light of several limitations. Firstly, our datasets are cross-sectional in nature. Therefore, we cannot draw firm conclusions regarding causality of any effects. Consequently, we cannot determine the long-term indirect effects of CoDG on self-esteem through NTB, nor the long-term indirect effect of CoDG on loneliness through NTB and self-esteem in men. Therefore, only longitudinal research would be able to verify whether a narrow CoDG is an antecedent of a high NTB, low self-esteem, and high loneliness in men, and further test the long-term effects of CoDG on self-esteem and loneliness. Secondly, our study may be limited in its assessment of the need to belong via the NTBS only. The construct of belonging need may be multifaceted, encompassing, but not limited to, a growth orientation (an approach-focused motive directing toward interpersonal actualization) and a deficit-reduction orientation (an avoidance-focused motive directing

toward interpersonal deficit reduction or repair) [75], but the NTBS primarily measures the avoidance-oriented aspect [76]. Future studies could enhance our understanding of the effects of CoDG on NTB by employing measurements that emphasize the approach-oriented aspect of NTB [76]. Finally, since the participants in the present study live in Western societies, the generalizability of the findings to participants coming from other cultural backgrounds, e.g., from East Asian societies, may be compromised. Individuals in East Asian societies emphasize the self as closely associated with others in their social groups, whereas individuals in Western societies view the self as independent [77]. Men in Western societies, who emphasize uniqueness in self-construal, associate the subjective feeling of whether being looked at or not by others with the belongingness need. However, one should be very conservative in assuming that the findings observed in Western men can be replicated in men who live in cultures emphasizing interdependence among individuals. As it was evidenced by the present study, the association of CoDG width with the belongingness need was not observed for women who construe the self as more interdependent and relational than men in Western societies.

## 6. Merits and conclusions

Firstly, the present research has notable theoretical and practical implications. Theoretically, it is the first to demonstrate that men's CoDG width, akin to a personality-like trait, may influence other psychological constructs such as the need to belong, self-esteem, and loneliness. In the real-world context, this research suggests that strategies for addressing low self-esteem and high loneliness in men should be focused on men's conception that they are valued and accepted by other people. Another merit of the study lies in examining the joint effects of CoDG and NTB on self-esteem and loneliness separately for men and women. We found that there was no association between CoDG width and the need to belong in women. We explained this by proposing that women's sociality orientation (i.e., close relationships) and mental representations of social groups (i.e., dyadic bonds) may result in the lack of association between CoDG and NTB. Admittedly, though, the precise nature of the difference between men and women remains uncertain. This underscores the importance for researchers in the field of social cognition to consider gender differences in their investigations, especially when investigating how the way women interpret social cues influences their need to belong, self-esteem, and loneliness.

Our study revealed two novel findings. Firstly, in men, a narrower CoDG, possibly leading to difficulties in forming and sustaining social relationships, predicted a higher NTB, lower self-esteem, and higher loneliness. Moreover, NTB played a significant role in how a narrow CoDG predicted low self-esteem and high loneliness. Specifically, a narrow CoDG predicted high loneliness through the serial mediation of NTB-self-esteem, while it predicted low self-esteem through the mediation of loneliness for those who had a high NTB. Last but not least, in women, the effects of the CoDG width on the other three psychological variables were not observed. This highlights the need for further research to develop a theoretical framework that considers how the CoDG width, in association with the need to belong, self-esteem, and loneliness, influences women's social interaction. One potential avenue is to identify key factors shaping the association between CoDG and socioemotional characteristics, such as factors related to observers (e.g., personality traits) and gazers (e.g., facial features or other information relevant to social desirability). Additionally, a network analysis approach, well-suited for examining the structures of complex relations, could help researchers explore the dynamics in a network involving women's eye contact perception, personality traits (e.g., Big Five traits), and the quality of their social interactions (e.g., general satisfaction in social relationships, satisfaction of romantic relationships, and friendships). The present study can be seen as a first step towards understanding these dynamics and encourages future research in this area.

## Supporting information

**S1 Table. Means and standard deviations of attractiveness ratings as a function of face identity and participants' gender, as well as the statistical results of pairwise comparisons between men and women for each identity's attractiveness ratings.**
(DOCX)

## Author contributions

**Conceptualization:** Manman Zhai, Jari K. Hietanen.

**Data curation:** Manman Zhai, Jari K. Hietanen.

**Formal analysis:** Manman Zhai.

**Funding acquisition:** Manman Zhai, Heini Saarimäki, Jari K. Hietanen.

**Investigation:** Manman Zhai.

**Methodology:** Manman Zhai, Heini Saarimäki, Jari K. Hietanen.

**Project administration:** Jari K. Hietanen.

**Resources:** Manman Zhai, Jari K. Hietanen.

**Software:** Manman Zhai.

**Supervision:** Jari K. Hietanen.

**Validation:** Manman Zhai, Heini Saarimäki, Jari K. Hietanen.

**Visualization:** Manman Zhai.

**Writing – original draft:** Manman Zhai.

**Writing – review & editing:** Manman Zhai, Heini Saarimäki, Jari K. Hietanen.

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
