## [Decision Letter · Decision Letter 0]

25 Dec 2024

PONE-D-24-25863The perception of eye contact is associated with the need to belong, self-esteem, and loneliness in men but not in womenPLOS ONE

Dear Dr. Zhai,

Thank you for submitting your manuscript to PLOS ONE. After careful consideration, we feel that it has merit but does not fully meet PLOS ONE’s publication criteria as it currently stands. Therefore, we invite you to submit a revised version of the manuscript that addresses the points raised during the review process.

Reviewer 1 Comments : Endorsed

Reviewer 2 comments :

1) Reviewer wants to know what the primary hypothesis of this study on the perception of eye contact and its associations is.

2) Reviewer wants to understand how the "cone of direct gaze" (CoDG) is defined in this study and why it is significant.

3) Reviewer wants to explore the main differences observed between men and women regarding CoDG and its psychological associations.

4) Reviewer wants to clarify what specific tasks participants completed during the eye contact judgment experiment.

5) Reviewer wants to understand how participants' levels of self-esteem, loneliness, and need to belong were measured.

6) Reviewer wants to examine why certain participants were excluded from the final analysis and how data quality was ensured.

7) Reviewer wants to investigate the relationship between CoDG width and loneliness among men.

8) Reviewer wants to analyze how NTB mediates the relationship between CoDG width and self-esteem.

9) Reviewer wants to know why CoDG was not associated with NTB, self-esteem, and loneliness among women.

10) Reviewer wants to understand how the findings explain the gender-specific effects of CoDG width on NTB and loneliness.

11) Reviewer wants to assess what theoretical explanations are proposed for why men and women experience different social dynamics based on CoDG.

12) Reviewer wants to identify the broader implications of this research for understanding social cognition and interpersonal relationships.

13) Reviewer wants to consider the limitations of using the NTBS questionnaire in studying the need to belong.

14) Reviewer wants to analyze how the study addresses potential cultural influences on the observed patterns.

15) Reviewer wants to know what avenues for future research are suggested by the findings, particularly regarding gender differences and CoDG

We look forward to receiving your revised manuscript.

Kind regards,

S Rehan Ahmad, PhD

Academic Editor

PLOS ONE

Journal Requirements:

**Comments to the Author**

1. Is the manuscript technically sound, and do the data support the conclusions?

Reviewer #1: Yes

Reviewer #2: Yes

2. Has the statistical analysis been performed appropriately and rigorously? 

Reviewer #1: Yes

Reviewer #2: Yes

3. Have the authors made all data underlying the findings in their manuscript fully available?

Reviewer #1: Yes

Reviewer #2: Yes

4. Is the manuscript presented in an intelligible fashion and written in standard English?

Reviewer #1: Yes

Reviewer #2: Yes

5. Review Comments to the Author

Reviewer #1: I think it is a very good attempt. The authors have presented all the details in a very intelligent and engaging fashion. What I would suggest though is that if the title can be amended slightly so that it is immproved. I would not like to see something in the title like "...in men but not in women".

Reviewer #2: 1) Reviewer wants to know what the primary hypothesis of this study on the perception of eye contact and its associations is.

2) Reviewer wants to understand how the "cone of direct gaze" (CoDG) is defined in this study and why it is significant.

3) Reviewer wants to explore the main differences observed between men and women regarding CoDG and its psychological associations.

4) Reviewer wants to clarify what specific tasks participants completed during the eye contact judgment experiment.

5) Reviewer wants to understand how participants' levels of self-esteem, loneliness, and need to belong were measured.

6) Reviewer wants to examine why certain participants were excluded from the final analysis and how data quality was ensured.

7) Reviewer wants to investigate the relationship between CoDG width and loneliness among men.

8) Reviewer wants to analyze how NTB mediates the relationship between CoDG width and self-esteem.

9) Reviewer wants to know why CoDG was not associated with NTB, self-esteem, and loneliness among women.

10) Reviewer wants to understand how the findings explain the gender-specific effects of CoDG width on NTB and loneliness.

11) Reviewer wants to assess what theoretical explanations are proposed for why men and women experience different social dynamics based on CoDG.

12) Reviewer wants to identify the broader implications of this research for understanding social cognition and interpersonal relationships.

13) Reviewer wants to consider the limitations of using the NTBS questionnaire in studying the need to belong.

14) Reviewer wants to analyze how the study addresses potential cultural influences on the observed patterns.

15) Reviewer wants to know what avenues for future research are suggested by the findings, particularly regarding gender differences and CoDG.

6. PLOS authors have the option to publish the peer review history of their article (what does this mean? ). If published, this will include your full peer review and any attached files.

**Do you want your identity to be public for this peer review?** For information about this choice, including consent withdrawal, please see our Privacy Policy .

Reviewer #1: No

Reviewer #2: No

---

## [Author Response · Author response to Decision Letter 1]

27 Jan 2025

Reviewer #1:

I think it is a very good attempt. The authors have presented all the details in a very intelligent and engaging fashion. What I would suggest though is that if the title can be amended slightly so that it is improved. I would not like to see something in the title like "...in men but not in women".

Thank you for the suggestion. As the study did not aim to make gender-based comparisons regarding the association between eye contact perception and individuals’ socioemotional characteristics, we revised the title to: “The perception of eye contact is associated with men’s need to belong, self-esteem, and loneliness”. This revised title highlights the key findings of our study.

Reviewer #2:

1) Reviewer wants to know what the primary hypothesis of this study on the perception of eye contact and its associations is.

In the revised version, we clarified the primary hypothesis of this study at the very beginning in section 1.3. “Overview of the present study” on Pages 8-9, Lines 170-178.

“The primary research question was whether CoDG predicts NTB, self-esteem, and loneliness. We hypothesized that individuals with a narrow CoDG may have a high NTB, low self-esteem, and high loneliness. In addition, based on various perspectives in the research literature, we also investigated whether CoDG predicts self-esteem and loneliness via NTB (serial mediation models, Figure 1a-1b; moderated mediation models, Figure 1c-1d). We first investigated the relationship between CoDG and NTB. Because previous research has shown that there are gender differences in the CoDG width [31,32] as well as in NTB strength [33,50,51], we also explored whether there were gender differences in NTB and CoDG in the present sample of participants. If that would be the case, we would analyse the results separately for men and women”.

2) Reviewer wants to understand how the "cone of direct gaze" (CoDG) is defined in this study and why it is significant.

We agree with the reviewer that it is important to explain how the “cone of direct gaze (CoDG)” was defined in the present study. Gamer and Hecht (2007) hypothesized that “there is a range of considerable width wherein a person feels looked at”, using the metaphor of a cone of direct gaze to describe this range of gaze directions perceived as eye contact by an observer. The definition of CoDG in this study aligns with this widely accepted descriptive concept.

However, CoDG can be measured using different methods, which may affect the estimation of its width (Linke & Horstmann, 2024). Thus, it is necessary to clarify the method used in the present study. Building on the work by Gibson and Pick (1963), participants were shown face images displaying varying gaze directions (ranging from 0˚ to 8˚ in an increment of 2˚) and asked to judge whether they were being looked at by responding with “yes” or “no” (Details of two alternative methods are available in Linke and Horstmann, 2024). Based on these dichotomous eye contact judgments and the continuous variation in gaze directions, a point of subjective equality (PSE)—the gaze direction with a 50% probability of being judged as a direct gaze or averted gaze—was calculated using a binary logistic regression model. The width of CoDG was then determined by multiplying PSE with two. Without this methodological clarification, readers might find the definition of CoDG unclear in Introduction. However, since the detailed calculation of CoDG width fits more appropriately in the method section, and the cone metaphor effectively conveys the concept of eye contact perception, we kept the definition brief in Introduction. In the revised version, we clarified this point and included a reminder directing readers to where CoDG calculation is explained in detail, in the manuscript on Page 3, Lines 54-56.

“This has been investigated by measuring the cone of direct gaze (CoDG), a range of gaze directions slightly deviating from the direct gaze which observers accept as eye contact [3] (see section Data analysis for how CoDG width was calculated)”.

3) Reviewer wants to explore the main differences observed between men and women regarding CoDG and its psychological associations.

The main difference is that, for men, CoDG width was correlated with NTB, self-esteem, and loneliness, whereas, for women, CoDG was not correlated with NTB, self-esteem, or loneliness. The specific values and significance are provided on Page 17, Lines 331-336 and in Table 2.

4) Reviewer wants to clarify what specific tasks participants completed during the eye contact judgment experiment.

As addressed in point 2), we presented face images displaying eleven gaze directions (left: 2˚, 4˚, 6˚, 8˚, 20˚; direct: 0˚; right: 2˚, 4˚, 6˚, 8˚, 20˚), which participants judged for making eye contact with them or not. Further details can be found on Page 11, Lines 226-238.

5) Reviewer wants to understand how participants' levels of self-esteem, loneliness, and need to belong were measured.

We measured the need to belong, self-esteem, and loneliness using the Need to Belong Scale (Leary et al., 2013), the Rosenberg Self-Esteem Scale (Rosenberg, 1965), and the UCLA Loneliness Scale (Russell, 1996), respectively. Participants completed these scales in the fixed order mentioned above. Instead of viewing all items at once, participants in our study were presented with one item each time, responding by pressing a number key. The items on each scale were presented in random order for each participant. Further details can be found on Pages 11-12, Lines 242-260.

6) Reviewer wants to examine why certain participants were excluded from the final analysis and how data quality was ensured.

To ensure data quality, we incorporated several attention check items into the experimental design and implemented strict data cleaning procedures.

Firstly, attention check items were included in the eye contact judgment task (16 attention-check trials among 160 test trials), the Need to Belong Scale (1 in 10), the Rosenberg Self-Esteem Scale (1 in 10), and the UCLA Loneliness Scale (2 in 20). Inspecting the responses to attention check items could reveal whether participants paid enough attention during the online tasks. In the eye contact judgment task, participants were excluded if their percentage of "yes" responses to catch trials (faces with gaze direction of 20˚) exceeded 50%. Similarly, in the questionnaire (composed of 40 items), participants were excluded if they failed more than two (out of four) attention check trials. In the revised version, this point was clarified on Pages 12-13, Lines 264-272:

“In the eye contact judgment task, the participant was excluded if the percentage of “yes” responses to catch trials (faces with gaze direction of 20˚) was more than 50%. One participant who gave 87.5% “yes” responses to these catch trials was excluded. In the questionnaire (composed of 40 items), participants were excluded if they failed more than two (out of four) attention check trials. No participant was excluded due to attention checks in the questionnaire (19 participants failed one attention check question, and 2 failed two). Thus, 399 participants’ data were analyzed subsequently. In the subsequent analysis of gaze cone for the 399 participants, those whose data did not meet the criteria for calculating CoDG width were excluded, even if they had passed the attention check questions among the three scales”.

For the final sample, we reported the average percentage of "yes" responses to catch trials in the eye contact judgment task on Page 14, Lines 294-297.

“We also checked these participants’ responses on attention check trials (16 trials of 20˚ gaze direction), and the average error rate on the catch trials was 0.46%. Two participants made 2 erroneous responses, twenty-three participants made 1, and the remaining 342 participants made no erroneous responses”.

Second, after removing attention check items, we applied a series of exclusion criteria when calculating CoDG width:

1 Pre-calculation exclusion: Participants were excluded if the percentage of “yes” responses to 8˚ gaze directions (leftward and rightward averted gaze) exceeded 50%, or if the percentage of “yes” responses to direct gaze (0˚) was below 50%. Additionally, participants who self-reported technical issues that could have affected their results were excluded.

2 post-calculation exclusion: The goodness of fit for binary logistic regression models used to calculate CoDG width was assessed by examining deviance residuals. Participants with deviance residuals exceeding their corresponding degrees of freedom, indicating a poor model fit, were excluded.

3 Manual inspection: CoDG width values were manually inspected for abnormalities, such as negative values.

Details about participant exclusions based on these criteria can be found on Pages 13-14, Lines 273-278 and Lines 288-298.

7) Reviewer wants to investigate the relationship between CoDG width and loneliness among men.

Among men, the results showed a negative correlation between CoDG width and loneliness level (Page 17, Line 333 and Table 2), suggesting that men with narrower CoDG reported higher loneliness. In social interactions, an averted gaze may signal avoidance-oriented motivation by the sender (Adam & Kleck, 2003, 2005) and activate a receiver’s motivation tendency of avoidance (Hietanen et al., 2008; Uusberg et al., 2015). Thus, a narrow CoDG, reflecting a tendency to interpret slight gaze deviations as averted gaze, may have adverse effects on the formation and maintenance of social bonds. This difficulty in initiating or sustaining desired social relationships can lead to social isolation (loneliness). A more detailed discussion of this point can be found on Pages 22-23, Lines 425-436.

8) Reviewer wants to analyze how NTB mediates the relationship between CoDG width and self-esteem.

In the serial mediation model, CoDG width predicted loneliness through the serial mediation of NTB and self-esteem, with NTB also mediating the association between CoDG width and self-esteem in men (Page 19, Lines 352-353). We suggested that a narrower CoDG may decrease opportunities for forming and maintaining relationships among men. The lack of regular contact or meaningful bonds with others cannot fully satisfy NTB. Further, a high NTB may increase concerns regarding relationship devaluation or lead to the eventuality of unsatisfying relationships, which is reflected in low self-esteem. A more detailed discussion is provided on Pages 24-25, Lines 467-482.

9) Reviewer wants to know why CoDG was not associated with NTB, self-esteem, and loneliness among women.

On Pages 26-27, Lines 499-521, we discuss why CoDG was not associated with NTB, self-esteem, and loneliness among women. We hypothesized that the joint effects of CoDG width and NTB would predict individuals’ self-esteem and loneliness levels. However, the absence of an association between CoDG and NTB may explain why CoDG width cannot serve as a predictor for self-esteem or loneliness. To address the reviewer’s question (also our own), we explored possible reasons for the gender difference in the CoDG-NTB association. Specifically, a negative correlation was observed between CoDG and NTB among men, whereas no such association was found for women. While people are fundamentally motivated to establish and sustain relationships regardless of gender, differences exist in how belongingness is pursued (Baumeister & Sommer, 1997) and how social relationships are mentally represented (Eck et al., 2017; Foels & Tomcho, 2009). Men may be oriented to large groups of shallow relationships and see social groups as overall categories (e.g., family or college). Therefore, CoDG width likely affects whether they can build broad social networks. In men, narrower CoDG together with the representation of social relationships as categories may make it more difficult in forming dyadic and meaningful social interactions. In contrast, women’s sociality is primarily oriented toward intimate and close relationships, and they see social relationships as dyadic bonds (e.g., sister or classmate). As such, judging nonsignificant others as not making eye contact does not affect their sense of belonging. Women just need to recognize potential dyadic relationships first and engage in meaningful interactions with specific individuals. Thus, their NTB can be met by social interactions of high quality, regardless of their CoDG width.

10) Reviewer wants to understand how the findings explain the gender-specific effects of CoDG width on NTB and loneliness.

If we understand this comment from the reviewer correctly, we want to raise two aspects in our response. On one hand, in this study, we did not find gender differences in loneliness. Although we observed gender differences in CoDG width and NTB strength, we cannot conclude that the gender difference in NTB is the effect of the gender difference in CoDG width. On the other hand, a negative correlation was observed between CoDG and NTB among men, whereas no such association was found for women. In men, narrower CoDG predicts higher loneliness through the link of higher NTB and lower self-esteem. However, in women, this study did not show the predictive effect of CoDG width on self-esteem or loneliness perhaps due to the absence of association between CoDG and NTB. Then, we can return to comment 9) to think about the gender difference in the CoDG-NTB association. As with how men’s CoDG width together with NTB exert influences on loneliness, the detailed discussion can be found on Pages 24-25, Lines 467-482.

11) Reviewer wants to assess what theoretical explanations are proposed for why men and women experience different social dynamics based on CoDG.

As addressed in our response to comment 9), with a more detailed discussion on Pages 26-27, Lines 499-521, we proposed that gender differences exist how belongingness is pursued (Baumeister & Sommer, 1997) and how social relationships are mentally represented (Eck et al., 2017; Foels & Tomcho, 2009). These theories may help explain why men and women experience different social dynamics based on CoDG. However, we want to emphasize that these interpretations are not exclusive and that there may be other theories accounting for our findings. Since this study aimed to tentatively explore whether there would be gender difference and what the gender difference would be in the use of CoDG width to predict socioemotional characteristics, we did not test any theories regarding the mechanisms underlying the social dynamics associated with CoDG.

12) Reviewer wants to identify the broader implications of this research for understanding social cognition and interpersonal relationships.

The broader theoretical and practical implications of this research can be found on Page 28, Lines 547-559.

13) Reviewer wants to consider the limitations of using the NTBS questionnaire in studying the need to belong.

The construct of the need to belong may be multifaceted, encompassing but not limited to a growth orientation (an approach-focused motive directing toward interpersonal actualization) and a deficit-reduction orientation (an avoidance-focused motive directing toward interpersonal deficit reduction or repair) (Lavigne et al., 2014). Notably, the NTBS primarily measures the avoidance-oriented aspect (Pillow et al., 2015). On Page 27, Lines 529-535, we acknowledged that our study might be limited by assessing the need to belong only via the NTBS questionnaire.

14) Reviewer wants to analyze how the study addresses potential cultural influences on the observed patterns.

On Pages 27-28, Lines 536-545, we tentatively analyzed the potential cultural influences on the observed patterns. However, since the study did not include a sample from another cultural background, we chose to avoid extensive speculation about cultural influences.

15) Reviewer wants to know what avenues for future research are suggested by the findings, particularly regarding gender differences and CoDG.

We appreciate the reviewer’s encouragement for further exploration. This study aimed to investigate how individuals’ eye contact perception predicts their socioemotional chara

---

## [Editor Report · Decision Letter 1]

12 Feb 2025

The perception of eye contact is associated with men's need to belong, self-esteem, and loneliness

PONE-D-24-25863R1

Dear Dr. Zhai

We’re pleased to inform you that your manuscript has been judged scientifically suitable for publication and will be formally accepted for publication once it meets all outstanding technical requirements.

Kind regards,

S Rehan Ahmad, PhD

Academic Editor

PLOS ONE

---

## [Editor Report · Acceptance letter]

PONE-D-24-25863R1

PLOS ONE

Dear Dr. Zhai,

I'm pleased to inform you that your manuscript has been deemed suitable for publication in PLOS ONE. Congratulations! Your manuscript is now being handed over to our production team.

Kind regards,

on behalf of

Dr. S Rehan Ahmad

Academic Editor

PLOS ONE